# Estimating the Volume of Nodules and Masses on Serial Chest Radiography Using a Deep-Learning-Based Automatic Detection Algorithm: A Preliminary Study

**DOI:** 10.3390/diagnostics13122060

**Published:** 2023-06-14

**Authors:** Chae Young Lim, Yoon Ki Cha, Myung Jin Chung, Subin Park, Soyoung Park, Jung Han Woo, Jong Hee Kim

**Affiliations:** 1Department of Radiology, Samsung Medical Center, Sungkyunkwan University School of Medicine, Seoul 06351, Republic of Korea; star05890@gmail.com (C.Y.L.); mj1.chung@samsung.com (M.J.C.);; 2Department of Health Sciences and Technology, SAIHST, Sungkyunkwan University, Suwon 16419, Republic of Korea

**Keywords:** pulmonary nodules, chest X-rays, deep-learning, convolutional neural networks, volume estimation

## Abstract

Background: The purpose of this study was to assess the volume of the pulmonary nodules and masses on serial chest X-rays (CXRs) from deep-learning-based automatic detection algorithm (DLAD)-based parameters. Methods: In a retrospective single-institutional study, 72 patients, who obtained serial CXRs (*n* = 147) for pulmonary nodules or masses with corresponding chest CT images as the reference standards, were included. A pre-trained DLAD based on a convolutional neural network was developed to detect and localize nodules using 13,710 radiographs and to calculate a localization map and the derived parameters (e.g., the area and mean probability value of pulmonary nodules) for each CXR, including serial follow-ups. For validation, reference 3D CT volumes were measured semi-automatically. Volume prediction models for pulmonary nodules were established through univariable or multivariable, and linear or non-linear regression analyses with the parameters. A polynomial regression analysis was performed as a method of a non-linear regression model. Results: Of the 147 CXRs and 208 nodules of 72 patients, the mean volume of nodules or masses was measured as 9.37 ± 11.69 cm^3^ (mean ± standard deviation). The area and CT volume demonstrated a linear correlation of moderate strength (i.e., R = 0.58, RMSE: 9449.9 mm^3^ m^3^ in a linear regression analysis). The area and mean probability values exhibited a strong linear correlation (R = 0.73). The volume prediction performance based on a multivariable regression model was best with a mean probability and unit-adjusted area (i.e., RMSE: 7975.6 mm^3^, the smallest among the other variable parameters). Conclusions: The prediction model with the area and the mean probability based on the DLAD showed a rather accurate quantitative estimation of pulmonary nodule or mass volume and the change in serial CXRs.

## 1. Introduction

In the last decade, deep-learning-based automatic detection (DLAD) algorithms have significantly surpassed conventional computer-aided detection (CAD) systems in a variety of tasks related to chest radiographs (CXR) [1,2,3,4]. These advanced algorithms have seen an increased adoption in clinical settings for detecting and classifying a wide range of abnormal pulmonary lesions such as nodules, masses, and COVID-19 pneumonia [5,6,7,8,9]. They provide invaluable assistance to physicians, particularly in cases where radiologists are unavailable or overburdened [6,7].

They also enhance the radiologists’ performances when used as a support [10,11,12,13]. In our study utilizing DLAD, the average sensitivity of radiologists improved from 65.1% to 70.3%, and the number of false-positive findings per radiograph decreased from 0.2 to 0.18 when the radiologists re-reviewed the radiographs with the aid of the DLAD software. Among the 12 radiologists, 104 of 2400 radiographs experienced positive changes (i.e., from a false-negative to a true-positive or from a false-positive to a true-negative) using the DLAD, while 56 of 2400 radiographs showed negative changes.

Despite their impressive capabilities, DLAD algorithms currently lack the ability to compare serial CXRs for assessing lesion size or severity, which is a crucial task performed by radiologists daily.

Convolutional neural networks (CNNs), a specialized form of deep-learning, have piqued interest among the researchers and developers of chest radiography algorithms [14]. Utilizing class activation maps (CAM) in conjunction with CNN models can significantly enhance the image interpretability and localization [15,16]. The CAM computes the probability function for a specific location (x, y) on an input image, generating probability maps that can help pinpoint key areas in the image [16].

The research hypothesis states that the higher the probability of a lesion’s existence, the more distinct and denser the lesion appears, potentially indicating a larger projected volume for the lesion. To test this hypothesis, regions of interest (ROIs) with probabilities exceeding a certain threshold were identified as abnormal on the probability map, and the area of the ROI was estimated for each image. By combining this area and probability value, which was calculated using the DLAD classification algorithm, researchers investigated the potential for estimating the volume of the target lesion.

The goal of this research is to apply these findings to the comparison of serial CXRs, allowing for a more comprehensive analysis of the lesion size and severity over time. This advancement could provide physicians with an additional layer of information, thus further enhancing their ability to diagnose and treat various pulmonary conditions. There has been no previous attempt, as shown in this study, to use DLAD for estimating the 3D volume from 2D chest X-rays and to utilize this for assessing changes in lesions. This novel approach may yield academic contributions to the field of deep-learning in medical imaging.

## 2. Materials and Methods

This study was approved by our institutional review board (IRB No. SMC 2021-07-075-002) and individual consent for this retrospective analysis was waived.

A pilot study was conducted to derive the parameters from the DLAD of the researcher’s institution, thereby reflecting the 3 kinds of interval changes (i.e., increased, decreased, and no change) of nodules or masses in the CXRs. The reference standards were the formal readings of each CXR.

To quantitatively analyze the interval change of the nodules or masses, the researchers measured the 3D CT volume of nodules and used this value as a reference standard to perform a linear and non-linear correlation analysis with parameters derived from the pilot study and established prediction models.

### 2.1. Patients

We retrospectively searched the CXR database of a single, large academic institution (Samsung Medical Center, Seoul, Republic of Korea) from September 2018 to December 2020 using the combination of search terms of “increase-”, “decrease-”, “grow-”, “mass-”, “nodule-”, and “solitary pulmonary nodule-”, which resulted in the identification of 383 patients with interval-changed nodules and masses. Among them, 72 patients with follow-up images with the X-ray scanners of the same manufacturers as the previous images were selected. In the pilot study, when predicting the interval change with the probability map of images taken by scanners from different manufacturers, the accuracy was identified as low.

The ALND has been trained for nodules or masses with a diameter of 4.5 cm or less. Patients with a mass of 4.5 cm or larger were excluded during the initial stage or follow-up scans. Among them, 117 patients with corresponding chest CT scans of both the initial and follow-up radiographs, obtained before or after 7 days from the day of each CXR, were included. Lastly, we reviewed the PACS images and included 72 patients and a total of 147 chest radiographs with the inclusion and exclusion criteria as described in Figure 1.

### 2.2. CAD System for CXR Interpretation

The Samsung Auto Lung Nodule Detection (ALND, version 1.00; Samsung Electronics, Suwon, South Korea) was used in the present study. This deep convolutional neural network (DCNN) was built upon a modified version of ResNet-50 [17], and trained to detect lung nodules on digital radiographs. The training set was 13,710 normal radiographs and 3500 lung nodule (benign or malignant)-containing radiographs, while a test/validation set was 640 normal radiographs and 1480 lung-nodule radiographs (a total of 5700 lung nodules; size: 7 mm ≤ x ≤ 45 mm) [4]. The software was commercially available in Europe and South Korea as of June 2019. For the results, a probability score of 15% was defined as the threshold for a binary classification between the negative and positive results.

The trained ALND generated continuous probability values between 0 and 1 for each chest radiograph, corresponding to the probability that there would be nodules on the chest radiograph. In addition, the DLAD produced continuous activation values ranging from 0 to 1 for each pixel of the images, derived from the class activation function of the neural network [2]. With the cutoff of a 0.2 probability value, we obtained a CAM for each CXR and calculated the preselected parameters from the regions of interest (ROIs), with up to 3 for each image. The five preselected parameters obtained from the pilot study were the area, weighted opacity-area product (WOAP), mean, median, and maximum probability value of each ROI.

### 2.3. Image Analysis and Definition of Reference Standards with Volume Measurements

Two radiologists (Y.K.C., a thoracic radiologist with 16 years of experience, and M.J.C., a thoracic radiologist with 30 years of experience) reviewed all chest radiographs by consensus. The researchers selected up to three index pulmonary nodules that met the definition of the Fleishner Society glossary of terms and evaluated the interval change of each lesion [18]. At the serial follow-up CXRs, interval changes were classified as an increase or decrease in the extent of nodules and masses. For evaluating the volumes of indexed nodules, we measured the nodule volume semi-automatically with the CT scanner-linked Aquarius iNtuition^®^ software (TeraRecon, Houston, TX, USA) for the nodules detected at the baseline and follow-up CT examinations, based on the region-growing segmentation. In the case of an incomplete segmentation, the radiologist obtained the volume using the manual correction method [19]. Usually, the manual correction method has been adopted for juxta-structural masses or nodules as the volumetry included a large component of the vasculature or pleura [20].

### 2.4. Statistical Analysis for Establishing a Prediction Model

Spearman correlation coefficient tests were performed to estimate the correlation coefficients between the parameters and reference 3D CT volume, and the in-between parameters. Volume prediction models for the pulmonary nodules were established through univariable or multivariable, and linear or non-linear regression analyses with the parameters. A polynomial regression analysis was performed as a method of the non-linear regression model. The researchers established the degree of regression based on the assumption that the volume would be proportional to the area raised to the power of 1.5 and then adjusted the units between the area and volume (Appendix A). The root mean square error (RMSE) was measured to compare the performance of the polynomial regression models established at variable degrees.

## 3. Results

### 3.1. Demographics and Chest Radiography Features

A total of 147 CXRs from 72 patients (i.e., 39 men [54%] and 33 women [46%]; mean age, 62 years; age range, 33–84 years) with pulmonary nodules were included in the study.

Of the 147 CXRs, a total of 208 nodules were included when we identified up to 3 nodules per image in a descending order of size. The mean volume of the included nodules was 9.4 cm^3^ (range: 0.11 to 70.4 cm^3^). Lung cancer was pathologically confirmed in 34 (47%) patients with or without intrapulmonary metastases. Twenty-nine (40%) patients were pathologically or clinically diagnosed with pulmonary metastases of other primary malignancies. Eight (11%) patients were pathologically or clinically diagnosed with benign conditions including pulmonary tuberculosis, granulomatosis polyangiitis, cryptogenic organizing pneumonia, and nontuberculous mycobacteria infection. One patient was pathologically confirmed with primary chondrosarcoma originating from the thoracic cage. Seventy-eight (53%) CXRs were obtained from the scanners manufactured by GE Healthcare and 69 (47%) CXRs were obtained by Samsung Electronics (Table 1).

### 3.2. Regression Analysis of Volume Prediction of Pulmonary Nodules or Masses

Figure 2 shows the univariable regression analysis to explore the correlation between the area over the probability of 0.2 at the CAM and 3D CT volume of the nodule or mass, the mean probability of the ROIs over the cutoff probability value of 0.2 and the volume, and the in-between area and mean probability.

The area and CT volume demonstrated a linear correlation of moderate strength (R = 0.58). A quadratic regression analysis among the area and CT volume showed a similar strength of correlation as the linear regression (R-square = 0.34). The area and mean probability of the ROIs exhibited a strong linear correlation (R = 0.73). A quadratic regression analysis among the area and probability also showed a similar correlation strength compared to the linear regression (R-square = 0.53). Between the mean probability of ROIs and the CT volume, negligible correlations were noted in both the linear and quadratic regression analysis (R = 0.22 and R-square = 0.05).

Figure 3 shows the result of fitting the area and detection probability to the 2D plane using a linear regression equation concerning the CT volume from various angles. The equation for a 2D plane was 14,886.637214 + (29.288656) × Area − (57,206.109259) × Mean probability = Volume. The CT volumes of some masses were relatively large; therefore, there were outliers that rose significantly on the plane (e.g., four cases over 40,000 mm^3^, annotated by the red circle in Figure 3c). As shown in the univariable regression analysis, there was a strong correlation between the area and detection probability; therefore, in the quadratic regression equation, the coefficients appear as plus and minus signs, respectively.

Figure 4 demonstrates the correlation between the polynomial regression models of the degrees 1 (linear), 1.5 (unit-adjusted), and 2 (quadratic), estimated by the root mean squared error (RMSE). The performance for predicting the 3D CT volume with the area was the best in the unit-adjusted regression model (Figure 4a, RMSE = 9395.73 mm^3^). The performance for predicting the 3D CT volume with the area and mean probability was also the best at the unit-adjusted multivariable regression model (Figure 4d, RMSE = 7975.55 mm^3^). The residual plot for the unit-adjusted multivariable regression model is in the Appendix A. The equation for this model was = 578,437.473048 + 52,661.249342 × (Area^0.5^) + (−4,508,167.245662) × (Mean probability^0.5^) + 826.73901 × Area + (−217,913.487756) × (Area^0.5^) × (Mean probability^0.5^) + 10,744,046.813175 × Mean probability + (−3.158884) × (Area^0.5^) + (−804.666435) × Area × (Mean probability^0.5^) + 191,918.949126 × (Area^0.5^) × Mean probability + (−7,729,391.963889) × (Mean probability^0.5^). An example of applying this prediction model is demonstrated in Figure 5.

## 4. Discussion

We analyzed 147 chest radiographs (CXRs) from 72 patients, identifying a total of 208 pulmonary nodules and found a moderate linear correlation between the area of the nodule and its CT volume, and a strong linear correlation between the area and the mean probability of the regions of interest (ROIs). The best performance for predicting the 3D CT volume was achieved with a unit-adjusted regression model, which utilized both the area and mean probability generated by the DLAD.

The choice of a 1.5-degree polynomial regression model was driven by two main considerations. First, we aimed to adjust the units from 2D (mm^2^) to 3D (mm^3^), which naturally suggested a 1.5-degree polynomial. Second, we sought to align the dimensions from the 2D CXRs to the 3D volume of the nodules or masses. The 1.5-degree polynomial regression model demonstrated the best performance in terms of RMSE, indicating that it provided the most accurate estimates of the volume of the pulmonary nodules or masses.

Several previous studies have compared differences in the nodule or mass size in serial chest CT scans using DLAD systems. For example, Xu et al. predicted the treatment response of locally-advanced lung cancer patients using a combined CNN and RNN model, which analyzed the changes in mass on serial chest CT scans obtained during pretreatment and follow-up after radiotherapy [21]. Coreline Ltd. (Seoul, Republic of Korea), a company specializing in DLAD technology, has developed a commercial system capable of automatically matching the same nodule from previous chest CT scans and follow-up scans, facilitating the immediate identification of nodule changes [22]. In our study, 87% of patients had lung cancer or pulmonary metastasis. CXR examinations offer a more cost-effective and accessible method for short-term follow-up compared to chest CT scans, enabling clinicians to quickly assess the disease progression in their patients.

High-performance segmentation models based on U-Net architectures have been reported, providing more precise localization maps [14]. Shimazaki et al. developed a multiclass CXR abnormality detection algorithm using a segmentation method to calculate the exact nodule volume and to enable its comparison in serial images, although the accuracy of comparing the serial images was not validated [17]. We opted not to use U-Net-based architectures for serial image comparison, as our primary goal was to efficiently extract features reflecting the interval changes in nodules rather than achieving high-performance segmentation.

We hypothesize that a probability map can provide information on the rough area of the lesion seen on the CXR, which is not definite; however, a probability map is not made for measuring the area of a lesion but for the probability of the lesion’s existence. We thought that since the lesion is more definite, the probability values would get higher and that this might reflect the higher volume of a true lesion. By creating the equations by area and mean probability, our prediction of the lesion volume for nodules and masses was rather well predicted from serial CXRs, as an increasing or decreasing in the size via the reference standard of a measured 3D volume from CT.

Although our ALND model was trained using consecutive CXRs from multiple radiography device manufacturers, our hypothesis did not hold up well when the manufacturer of the serial CXRs changed [4]. Consequently, we developed a prediction model using serial CXRs from the same manufacturer.

Our approach for estimating the volume change using CAM proved effective, even though it was originally developed for multiclass discriminative models, unlike our binary classifiers [16]. It is anticipated that applying this technique to a multiclass discriminative model would yield a robust performance, thus extending its applicability beyond binary classifiers. Other multiclass discriminators have been introduced for CXR abnormalities, such as consolidation, interstitial opacity, pneumothorax, and pleural effusion [23]. Incorporating CAM into these models could lay the groundwork for future studies, including a pneumonia volume estimation for consolidation, ultimately assisting clinicians in making informed therapeutic decisions.

In addition to comparing lesions themselves, accurately identifying lesions across serial imaging follow-ups is a critical aspect of the monitoring process. Li et al. developed the concept of a ResNet-driven Siamese tracker for the visual tracking of images [24,25]. Cai et al. introduced the concept of a deep lesion tracker (DLT) for monitoring lesions in 4D longitudinal CT imaging studies. This method incorporated both image and anatomical considerations through self-supervised learning based on a Siamese neural network. The DLT reported an average Euclidean center error of 7 mm and improved the tumor monitoring workflow in assessing lesion treatment responses while maintaining an accuracy of 85%, which is comparable to manual lesion tracking [26]. Chen et al. introduced a transformer tracking method, known as TransT, which is built upon a Siamese-like feature extraction backbone, an attention-based fusion mechanism, and a classification and regression head. This approach was designed to address the loss of semantic information that occurs during the correlation operation in traditional tracking algorithms. By incorporating attention mechanisms into the tracking process, TransT effectively enhances the performance of lesion tracking. When tested on large-scale datasets such as LaSOT, TrackingNet, and GOT-10 k, TransT demonstrated a superior performance compared to previously reported lesion tracking algorithms that did not utilize attention mechanisms. Furthermore, Tang et al. proposed a transformer lesion tracker that employed a cross attention-based transformer (CAT) to capture and combine both global and local information for enhanced feature extraction. They also introduced a registration-based anatomical attention module to provide anatomical information to the CAT, enabling it to focus on relevant feature knowledge. A sparse selection strategy (SSS) is presented for selecting the features and reducing the memory footprint in transformer training. This approach improved the average Euclidean center error by at least 14.3% (6 mm vs. 7 mm) compared to the DLT when implemented on a test dataset of CT images, which the DLT also used [27]. Incorporating state-of-the-art tracking methods with the concept of our lesion volume change prediction model could significantly enhance the efficiency and accuracy of radiologists’ daily workflows. Furthermore, these techniques may not necessitate extensive training datasets, as the Siamese network for lesion tracking was initially developed for one-shot learning, and our prediction model was also based on a relatively small number of images with a pre-trained DLAD. By combining advanced lesion tracking technology with our volume change prediction model, we can potentially streamline the monitoring process and alleviate the workload for radiologists.

There are several limitations to this study. Firstly, as a single-center study, it included a limited number of patients. Secondly, we did not perform an internal validation of the suggested prediction model, thereby limiting our ability to check the final model’s fitness and adjust equation degrees and coefficients. Moreover, the accuracy of CAM for localization remains questionable. Additionally, more recent CAM methods have been proposed, exhibiting an improved performance [28,29]. For instance, Grad-CAM, a popular method for explaining the interpretability of DLAD, generates localization maps without additional training for CAM weights by utilizing the gradients calculated during backpropagation [30]. Specific clinical applications of this prediction model have not been identified. A decision curve analysis with the appropriate setting of clinical outcomes and interventions such as timely CT examination could further be evaluated [31]. Additionally, further studies for a multivariable prediction model of disease progression by adding clinical factors may help with clinicians’ decision making. Lastly, unlike detection and segmentation algorithms, which have state-of-the-art methodologies and appropriate metrics including a mean average precision, there are no reported state-of-the-art methodologies to be compared with our study.

## 5. Conclusions

In conclusion, the prediction model with the area and the mean probability of CAM showed the proper estimation of interval changes in pulmonary nodules or masses on the initial follow-up of serial CXRs with a quantitative volume estimation, based on commercially-available DLAD detecting pulmonary nodules and masses.

## Figures and Tables

**Figure 1 diagnostics-13-02060-f001:**
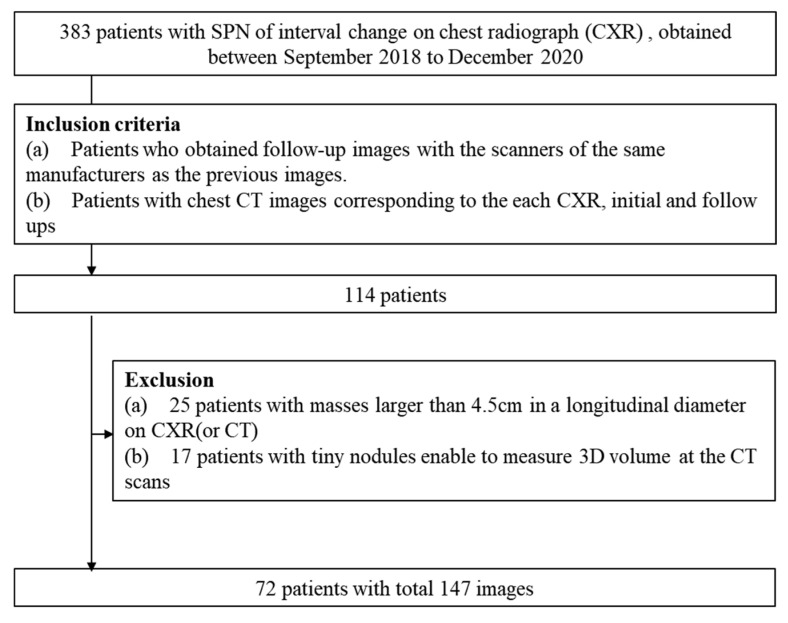
Study flowcharts.

**Figure 2 diagnostics-13-02060-f002:**
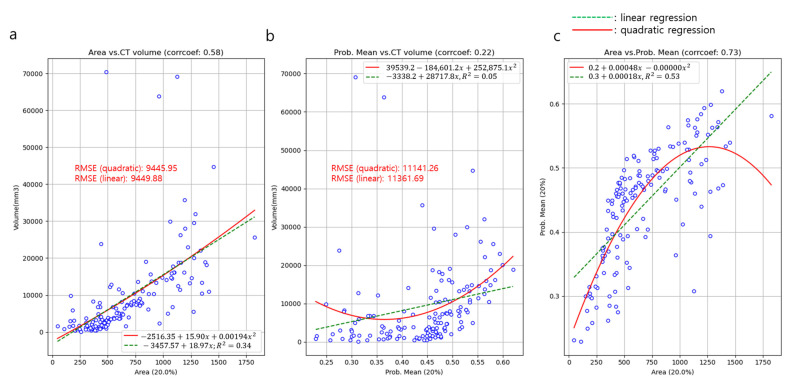
Univariable regression analysis to explore the correlation between the area over the probability of 0.2 (Area) at the CAM and 3D volume of the mass or nodule (**a**), mean probability (Prob. Mean) of the ROIs over the cutoff probability value of 0.2 and volume (**b**), and the in-between area and mean probability (**c**). (Note: regression model was not adjusted for age and sex).

**Figure 3 diagnostics-13-02060-f003:**
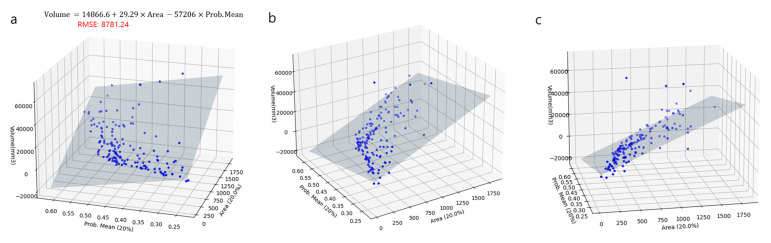
The two-dimensional plane fitted to the three-dimensional linear regression function graphs of area and detection probability for the CT volume at various angles (**a**–**c**). Four outliers over the CT volume of 40 cm^3^ are annotated by the red circle (**c**). (Note: Area = area over the probability of 0.2 at the class activation map, Prob.Mean = mean probability of the ROIs over the cutoff probability value of 0.2 in the class activation map).

**Figure 4 diagnostics-13-02060-f004:**
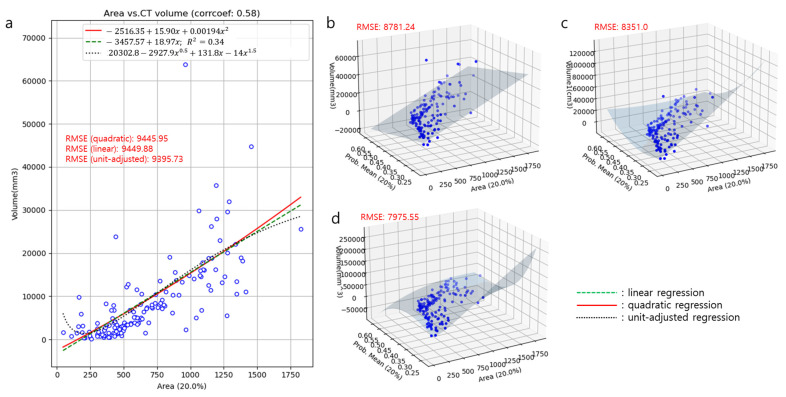
Correlation between the polynomial regression models of degrees 1, 1.5, and 2 estimated by RMSE. Comparison of performances for predicting 3D CT volume with the area among degree 1 (linear), 1.5 (unit-adjusted), and 2 (quadratic) univariate regression model (**a**). Comparison of performances for predicting 3D CT volume with the area and mean probability among degree 1 (linear), 1.5 (unit-adjusted), and 2 (quadratic) multivariable regression model (**b**–**d**). (RMSE = root mean square error).

**Figure 5 diagnostics-13-02060-f005:**
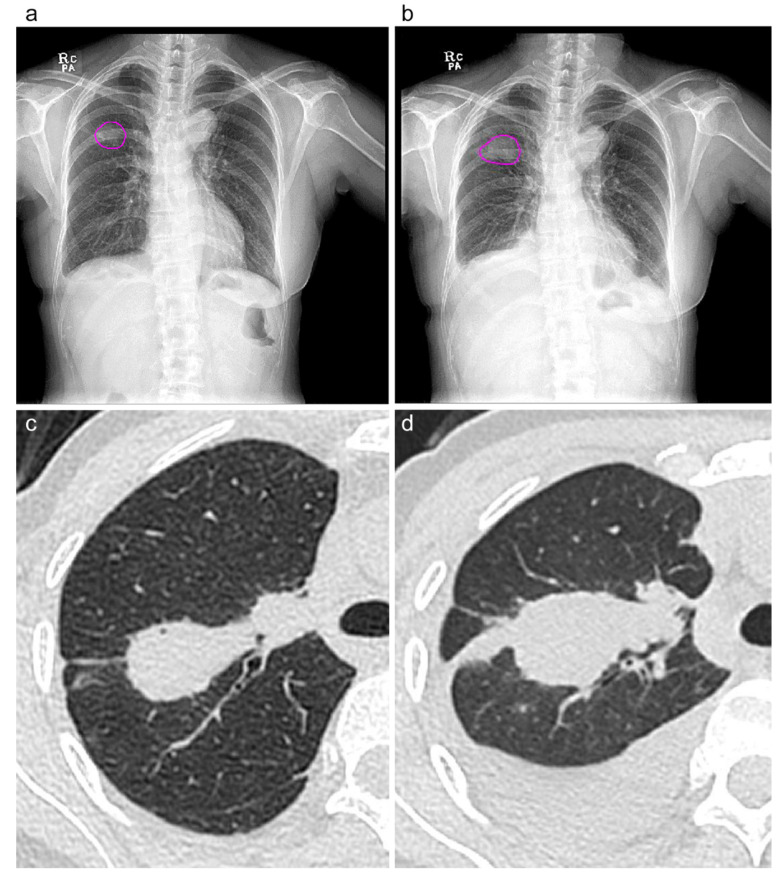
A 49-year-old female with breast cancer (pulmonary metastasis in RUL). In the initial and 2-month follow-up CXR (**a**,**b**), ALND-detected and -localized lesion nodule. Pink circles indicate the area, defined as the ROI over the 0.2 probability cutoff at the class activation channel, which is up-sampled and overlapped with the original image. The measured area and mean probability of each image were increased from 761.6 mm^2^ (7.62 cm^2^) and 0.514 (**a**) to 1079.6 mm^2^ (10.80 cm^2^) and 0.529 (**b**). Corresponding CT scans (**c**,**d**) within a week before or after the serial CXRs (**a**,**b**), the measured volume of the masses increased from 7.95 cm^3^ (**c**) to 14.50 cm^3^. The predicted volume of the 1.5-degree multivariable regression model was also increased from 8.50 cm^3^ to 16.35 cm^3^ (**d**).

**Table 1 diagnostics-13-02060-t001:** Patient information and nodule characteristics.

Characteristic	Number (Percentage)
Patient information	
No. of chest radiographs	147
Patient with nodules	72
No. of men	39 (54.17%)
No. of women	33 (45.83%)
Mean age	62.00 ± 11.74
Nodule information	
Total no. of nodules	208
No. of nodules per chest radiograph	
One nodule	104
Two nodules	25
Three nodules	18
Disease	
Primary lung cancer	34 (47.2%)
Pulmonary metastasis	29 (40.28%)
Benign	8 (11.11%)
Other primary malignancies in the thorax	1 (1.39%)
Mean nodule volume (cm^3^)	9.37 ± 11.69 *
Manufacturer	
GE Healthcare	78 (53.06%)
Samsung Electronics	69 (46.94%)

* mean ± standard deviation.

## Data Availability

The datasets generated during this study are not publicly available due to privacy and ethical considerations; however, anonymized data can be provided by the corresponding author upon reasonable request and with the approval of the ethics committee. Researchers interested in accessing the data should contact the corresponding author for further information.

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
