# Peer review of "Estimating the Volume of Nodules and Masses on Serial Chest Radiography Using a Deep-Learning-Based Automatic Detection Algorithm: A Preliminary Study"

_diagnostics, 2023, doi:10.3390/diagnostics13122060_

Round 1

Reviewer 1 Report

The paper demonstrates a scientific approach in the field of diagnostics, particularly in the context of pulmonary nodules or masses. The authors have effectively applied deep learning and computer-aided diagnosis techniques to develop a prediction model for estimating the volume and monitoring the interval changes of these abnormalities. The study is conducted as a single-centre study with a limited number of patients, which may restrict the generalizability of the findings. Internal validation of the prediction model is lacking, and the fitness of the final model and the adjustment of equation coefficients could not be evaluated. Additionally, the accuracy of the CAM (Class Activation Map) for localization is questionable, and newer methods with better performance have been suggested. Despite these limitations, the study's scientific soundness is apparent in its utilization of DLAD technology and its attempt to address the challenge of volume estimation in pulmonary nodules using CXR images.

It is worth noting that the paper lacks a separate conclusion section. Instead, the concluding remarks are incorporated within the discussion section. While this approach can be acceptable, a dedicated conclusion section can provide a succinct summary of the key findings and their implications. This would help readers quickly grasp the main outcomes of the study. Furthermore, the results section ends rather abruptly, transitioning directly into the discussion. It would be beneficial to have a more structured and conclusive ending to the results section, summarizing the main findings before moving on to the interpretation and implications of the discussion. The paper demonstrates a good incorporation of figures and tables within the text, enhancing the understanding of the research findings. Figures are appropriately referenced and discussed, providing visual representations of the regression analyses and prediction models. These figures aid in the interpretation of the results and strengthen the clarity of the presented data. Additionally, the references incorporated are well-established and up-to-date and support the research. 

Reviewer 2 Report

This retrospective study aimed to assess the volume of pulmonary nodules and masses on serial chest X-rays using a deep-learning-based automatic detection algorithm (DLAD). The study included 72 patients who underwent serial chest X-rays (147 in total) for pulmonary nodules or masses, with corresponding chest CT images as reference standards. A pre-trained DLAD based on a convolutional neural network was developed to detect and localize nodules and calculate various parameters such as the area and mean probability value of the nodules for each X-ray, including serial follow-ups.

The authors should address certain key issues and provide justifications to improve the manuscript. Additionally, there are a few concerns that need to be resolved.

1-     How were the interval changes of nodules or masses evaluated in the serial follow-up CXRs? Were there any specific criteria or guidelines used to classify the changes as an increase or decrease in extent?

2-      The main contributions need to be stated in the Introduction section

3-      In the related work, it is suggested to include the following research which utilized deep learning for extracting complex features: 

-       Liang CH, Liu YC, Wu MT, Garcia-Castro F, Alberich-Bayarri A, Wu FZ. Identifying pulmonary nodules or masses on chest radiography using deep learning: external validation and strategies to improve clinical practice. Clin Radiol. 2020 Jan;75(1):38-45. doi: 10.1016/j.crad.2019.08.005. Epub 2019 Sep 11. PMID: 31521323.

-       Abdussalam Elhanashi, Duncan Lowe, Sergio Saponara, Yashar Moshfeghi, "Deep learning techniques to identify and classify COVID-19 abnormalities on chest x-ray images," Proc. SPIE 12102, Real-Time Image Processing and Deep Learning 2022, 1210204 (27 May 2022); https://doi.org/10.1117/12.2618762.

4-      Did the prediction model based on the DLAD parameters accurately estimate the volume of pulmonary nodules or masses on serial CXRs?

5-      Can you provide more information about the polynomial regression analysis used as a method of the non-linear regression model? What were the different degrees of regression considered?

6-      The proposed work should be compared to the state of the art methodologies in terms of mean average precision obtain from this research

Minor proofreading is required 

Round 2

Reviewer 2 Report

Thanks to the authors for updating the manuscript accordingly.